# Effects of Exogenous Caffeic Acid, L-Phenylalanine and NaCl Treatments on Main Active Components Content and *In Vitro* Digestion of Germinated Tartary Buckwheat

**DOI:** 10.3390/foods11223682

**Published:** 2022-11-17

**Authors:** Wenping Peng, Nan Wang, Shunmin Wang, Junzhen Wang, Yulu Dong

**Affiliations:** 1College of Biological and Food Engineering, Anhui Polytechnic University, Wuhu 241000, China; 2Academy of Agricultural Science Liang Shan, Liangshan 615000, China

**Keywords:** Tartary buckwheat, caffeic acid, L-phenylalanine, NaCl, germination, *in vitro* digestion

## Abstract

Germination is an effective method for improving the nutritional value of Tartary buckwheat (TB). The effects of exogenous additive treatments (caffeic acid (CA), L-phenylalanine (L-Phe), NaCl) on germination, main active component contents and antioxidant activities before and after *in vitro* digestion of germinated TB were investigated. Compared with the natural growth group, the T_4_ group (CA 17 mg/L, L-Phe 2.7 mmol/L, NaCl 2.7 mmol/L) treatment increased the germination rate (67.50%), sprout length, reducing sugar (53.05%), total flavonoid (18.36%) and total phenolic (20.96%) content, and antioxidant capacity of TB. In addition, exogenous additives treatment induced the consumption of a lot of nutrients during seed germination, resulting in a decrease in the content of soluble protein and soluble sugar. The stress degree of natural germination on seeds was higher than that of low concentrations of exogenous additives, resulting in an increase in malondialdehyde content. *In vitro* digestion leads to a decrease in phenolics content and antioxidant capacity, which can be alleviated by exogenous treatment. The results showed that treatment with exogenous additives was a good method to increase the nutritional value of germinated TB, which provided a theoretical basis for screening suitable growth conditions for flavonoid enrichment.

## 1. Introduction

Buckwheat has a short harvest cycle, and some cultivars are more tolerant to adverse environmental conditions than several traditional crops. Because of these characteristics, buckwheat is planted in large quantities in some poor areas [1]. However, the poor yield and low germination rate of buckwheat make it a research goal of breeding experts to improve the germination characteristics of buckwheat [2]. Tartary buckwheat (TB) (*Fagopyrum tataricum* (L.) Gaertn.) is a plant of *Polygonaceae* and buckwheat. It is more attractive than common buckwheat because it is rich in flavonoids, vitamins, trace elements and various essential amino acids with antioxidant activity [3].

Germination has been recognized as an environmentally friendly, cheap and effective technology to improve the nutritional value of grains and the content of bioactive compounds [4,5]. A large number of studies have shown that the content of phytochemicals increases with germination [6,7]. Environmental growth conditions, such as NaCl stress, have been reported to increase levels of carotenoids, phenolic compounds and antioxidant activity in buckwheat sprouts [8]. Some the endogenous metabolites from plants, such as caffeic acid (CA) and L-phenylalanine (L-Phe), have also been used as additives to promote the accumulation of active substances. L-Phe is the initiator of the phenylpropane metabolic pathway, and the increase in its content directly affects the synthesis of polyphenols [9]. Previous studies have shown that L-Phe can increase the contents of total flavonoid (TF), total phenolic (TP), soluble sugars and reducing sugars after germination [10]. CA is not only the precursor of chlorogenic acid, but also the precursor of lignin and coumarin, which are involved in the synthesis of phenolics. In addition, it was reported that when CA and epicatechin were injected into apples, CA could activate the activities of peroxidase, multiple oxidases and phenylalanine ammonia-lyase, and increase the contents of flavonoids, TP and lignin [11]. In the process of seed germination, metabolism is exuberant, various enzymes are activated, different additives are converted to each other, nutrients are independent of each other and have certain correlations, and they work together to promote seed germination and provide energy for it. Therefore, it is of great significance to study the nutrients of TB during germination under the action of exogenous additives.

At present, there are many models for evaluating the bioaccessibility of bioactive additives. From these models, the *in vitro* digestion process has the advantages of high speed, low cost, safety and no ethical restrictions [12]. In recent years, simulated oral and gastrointestinal digestion has been widely used in many fields of food and nutrition science [13]. *In vitro* digestion procedures simulating stomach and small intestinal digestion have been successfully applied to various food substrates to analyze different antioxidant compounds, such as carotenoids, phenolics, vitamins, etc. [14]. In addition, gastrointestinal digestion or various food processing could promote the release of bioactive peptides from natural proteins, which in turn exert a wide range of biological functions, including antioxidant, anti-inflammatory, antibacterial, antithrombotic and blood pressure lowering functions such as inhibition of angiotensin-converting enzyme [15]. However, no research has been done to analyze the bioaccessibility and antioxidant capacity of phenolic compounds before and after *in vitro* digestion of germinated TB.

In this study, exogenous CA, L-Phe and NaCl were used to treat TB separately and cooperatively. The germination rate, sprout length, contents of main bioactive components before and after germination, TF, TP and antioxidant capacity before and after *in vitro* digestion were determined to select the most valuable exogenous treatment conditions, which provided a new research idea and direction for TB sprouts’ flavonoids enrichment.

## 2. Materials and Methods

### 2.1. Materials

TB seeds were selected from “Sichuan buckwheat No. 1”, provided by Xichang Institute of Agricultural Science (Sichuan, China). All reagents were analytically pure and purchased from the Sinopharm Chemical Reagent (Shanghai, China).

### 2.2. Germination Conditions

TB seeds were germinated according to the method described by Wang Simeng et al. [16] with minor modifications. Briefly, TB seeds were rinsed, immersed in distilled water and soaked at 25 °C for 4 h. After absorbing the soaking water, the seeds were uniformly sowed in trays (8 g per tray) to germinate with different cultivating solutions: T_1_, CA solution; T_2_, L-Phe solution; T_3_, NaCl solution; T_4_ and T_5_, mixed solution with CA, L-Phe and NaCl. The preparation concentrations of different treatment groups are shown in Table 1.

The seeds were germinated for 7 days in incubators (Heheng Instrument and Equipment, Shanghai, China) under the conditions of a constant temperature of 25 °C (humidity 80 ± 5%) and darkness. TB sprout samples were carefully collected after 3, 5 and 7 days of germination, removed from surface water, and then immediately stored at −80 °C for further phytochemical analysis. Fifty plant sprouts were taken from each treatment and set as a sampling group to measure the length and germination rate.

### 2.3. Measurement of Soluble Protein Content

An amount of 0.2 g fresh TB sprouts were ground into homogenate and transferred to a 10 mL volumetric flask to fix the volume and shake well. The 3 mL homogenate was centrifuged (Koki Holdings, Tokyo, Japan) at 8000 rpm for 10 min, and the supernatant was protein extract. The content of soluble protein in TB sprouts was determined by Coomassie brilliant blue G-250 staining according to the AOAC method [17]. The 0.1 mL protein extract was mixed with 0.9 mL distilled water and 5 mL Coomassie brilliant blue solution. The mixture was incubated at room temperature for 2 min and its absorbance (UV-5500PC, Shanghai Metash Instruments Co., Ltd, Shanghai, China) was determined at 595 nm. And bovine serum protein was used as a standard substance to make a standard curve (*y* = 0.0072*x* + 0.0553, *R*^2^ = 0.9976). The soluble protein content of the samples was expressed as the amount of bovine serum protein contained per 100 g dry weight (DW) of the TB sprout sample (g/100 g DW).

### 2.4. Measurement of Soluble Sugar and Reducing Sugar Contents

An amount of 0.5 g TB sprout powder ground after being frozen in a vacuum freeze dryer (Beijing Songyuan Huaxing Technology Develop, Beijing, China) was mixed with 6 mL 80% ethanol solution and extracted in a water bath at 80 °C for 30 min. After cooling to room temperature, the mixture was centrifuged at 5000 rpm for 10 min. The supernatant was placed in a 25 mL volumetric flask. Then, 5–6 mL 80% ethanol solution was added to the precipitation and extracted twice. All the obtained supernatant was merged into the volumetric flask and the volume fixed to the scale. An amount of 2 mL of the extract was dried in a boiling water bath in a centrifuge tube, then 10 mL of distilled water was added and stirred fully to dissolve the sugar. The mixture was centrifuged at 8000 rpm for 10 min and the supernatant was collected. The supernatant was used for the determination of soluble sugar and reducing sugar.

The content of soluble sugar in TB sprouts was determined by anthrone colorimetry [18]. An amount of 2 mL supernatant diluted 10 times was mixed with 5 mL anthrone-sulfuric acid reagent in an ice water bath and reacted in a boiling water bath at 100 °C for 10 min. Subsequently, the mixture was cooled to room temperature in ice water and the absorbance was determined at 620 nm. The standard curve of the glucose standard solution was drawn as *y* = 0.0074*x* + 0.0883 (*R*^2^ = 0.9998). The soluble sugar content in the sample was expressed by the glucose content per 100 g DW of the TB sprout sample (g/100 g DW).

The content of reducing sugar in TB sprouts was determined by the 3,5-dinitrosalicylic acid method according to the AOAC method [17]. An amount of 2 mL supernatant was mixed with 2 mL 3,5-dinitrosalicylic acid, and reacted in a boiling water bath for 5 min. After cooling to room temperature with running water, distilled water was added to fix the volume to 20 mL. Finally, the absorbance was determined at 540 nm. The standard curve of the glucose standard solution was drawn as *y* = 0.2980*x* − 0.0176 (*R*^2^ = 0.9989). The reducing sugar content in the sample was expressed by the glucose content per 100 g DW of the TB sprouts sample (g/100 g DW).

### 2.5. Measurement of Malondialdehyde (MDA) Content

The content of MDA was determined by the thiobarbituric acid method [19]. An amount of 0.3 g fresh TB sprouts was mixed with 2 mL 0.05 mol/L phosphate buffer and ground into homogenate in an ice bath. The homogenate was moved into the test tube, and 5 mL of 0.5% thiobarbituric acid solution was mixed into the test tube to react for 10 min in a boiling water bath, then immediately cooled in an ice water bath. After centrifuging at 5000 rpm for 15 min, the volume of the supernatant was recorded, and the absorbance was determined at 532, 660, and 450 nm. The determination result of MDA content was expressed as the amount of mmol per g DW of the TB sprout sample (mmol/g DW).

### 2.6. Measurement of TP and TF Contents

An amount of 0.2 g TB seeds and sprouts (3, 5 and 7 days) were ultrasonically extracted with 4 mL 80% methanol for 30 min, respectively. Then, the solutions were centrifuged at 8000 rpm for 10 min and the supernatant was collected. TP content was assayed by Folin-Ciocalteu colorimetric according to the AOAC method [17]. The diluted sample solution of 0.1 μL was added to 1 mL of Folin-Ciocalteu reagents. After incubation for 5 min, a 7.5% sodium carbonate solution of 1 mL was added. The mixture was incubated at room temperature for 30 min in the dark and its absorbance was determined at 765 nm. Results were expressed as mg gallic acid equivalents (GAE) per g DW of seeds or sprouts (mg GAE/g).

TF content was determined as described by Ji et al. [20] with slight modifications. An amount of 0.5 mL supernatant was mixed with 0.5 mL 5% sodium nitrite and incubated for 6 min, and then 0.5 mL of 10% aluminum nitrate was added. After 6 min of incubation, 4 mL 4% sodium hydroxide was added for further 12-min incubation. Then, the absorbance of the mixture was measured at 502 nm. The TF content was expressed as g of rutin equivalents (RE) per 100 g DW of seeds or sprouts (g RE/100 g).

### 2.7. Measurement of Antioxidant Capacities

The antioxidant capacities were determined by DPPH, ABTS radical scavenging capacity assay and ferric reducing antioxidant potential (FRAP). The DPPH radical scavenging capacity assay, described by Živković et al. [21], was conducted with minor modifications. An amount of 4.8 mL of 0.1 mg/mL DPPH solution was added to 0.2 mL of the appropriately-diluted extracted sample and the mixture was allowed to react in the dark at room temperature for 30 min. Then, the absorbance of the solution was monitored at 517 nm and anhydrous ethanol was used as a reference. The control consisted of 0.2 mL methanol and 4.8 mL DPPH solution.

The ABTS radical scavenging capacity assay, described by Živković et al. [21], was conducted with minor modifications. An amount of 4 mL of 7 mmol/L ABTS solution was added to 0.2 mL of the appropriately-diluted extracted sample and the mixture was allowed to react in the dark at room temperature for 30 min. Then, the absorbance of the solution was monitored at 734 nm and anhydrous ethanol was used as a reference. The control consisted of 0.2 mL methanol and 4 mL ABTS solution.

The FRAP was assayed according to the method by Yuan et al. [22] with minor modifications. The FRAP working solution of 3 mL (0.3 M acetate buffer, 0.02 M ferric chloride, 0.01 M 2,4,6-tripyridyl-S-triazine in 0.04 M hydrochloric acid at the ratio of 10:1:1) was added to the diluted extracted sample (100 μL). After the mixture was incubated at 37 °C for 30 min, the absorbance was recorded at 593 nm.

DPPH radical scavenging capacity, ABTS radical scavenging capacity and FRAP values were calculated by using 6-hydroxy-2,5,7,8-tetramethylchroman-2-carboxylic acid (Trolox) as the standard curve and expressed as μmol Trolox per g DW of seeds or sprouts (μmol TE/g).

### 2.8. In Vitro Digestion

The *in vitro* digestion was performed as described by Lv et al. [23] and Ortega-Vidal et al. [24]. All the simulated digestion steps were carried out in a constant temperature water bath shaker at 37 °C (150 r/min), under constant shaking in the dark.

For the oral phase, 1 g lyophilized seed powder and equivalent sprout powder were added to 10 mL of 0.9% NaCl solution. The suspension was added to 5 mL of saliva-simulated fluid for 5 min. For the gastric phase, 1 g freeze-dried seed or equivalent sprout was mixed with 10 mL 0.9% NaCl solution, then 5 mL saliva-simulated fluid was added and mixed for 5 min, and then 10 mL gastric-simulated fluid was added to the mixture for 2 h (adjusted pH 2.0 with 1 M HCl solution). For the intestinal phase, 1 g freeze-dried seed or equivalent sprout was mixed with 10 mL 0.9% NaCl solution, then 5 mL saliva-simulated fluid was added and mixed for 5 min, then 10 mL gastric-simulated fluid was added to the mixture for 2 h (adjusted pH 2.0 with 1 M HCl), and then 6 mL of simulated intestinal fluid was added to the mixture for another 2 h (adjusted pH 6.9 with 1 M NaHCO_3_). The above three solutions were centrifuged, and the supernatant was frozen at −80 °C for subsequent analysis. The *in vitro* digestion process for all samples was conducted in triplicate.

An index of bioaccessibility was adopted to reflect the effects of *in vitro* digestion on selected bioactive compounds. Bioaccessibility is defined as the percentage of these compounds that are solubilized in the gastrointestinal tract and becomes available for absorption [25], which can be calculated following Equation (1).
(1)$$Bioaccessibility\;\left(\%\right)=\frac{A}{B}\times 100$$
where: *A* is the content of the compounds (TP, TF or antioxidant capacity) released after intestinal digestion, and *B* is the content of the compounds (TP, TF or antioxidant capacity) in the food matrix (before digestion).

### 2.9. Statistical Analysis

All experimental data are presented as means ± standard deviation (SD) of independent replicates in triplicate. SPSS 22.0 (International Business Machines Corporation, Armonk, NY, USA)was used for significance analysis and correlation analysis, and Duncan’s multiple comparison method was used for variance analysis. The drawing was made by SigmaPlot 14.0 software (Systat Software, Inc., San Jose, CA, USA).

## 3. Results

### 3.1. Effect of Exogenous Additives on the Growth of Treated Sprouts

We set the seed germination rate of 1 day of germination as the initial germination rate, and the seed germination rate of 7 days as the final germination rate. As shown in Figure 1A, under different treatments, the initial germination rate of seeds was significantly higher than that of the CK group, and the highest initial germination rate of the T_4_ group was 67.00%, which was 67.50% higher than that of the CK group. The final germination rate of seeds in all treatment groups reached 99.00%, which was still the lowest in the CK group, at 90.33%, indicating that the treatment with an appropriate concentration of exogenous solution had a better effect on promoting the germination of TB seeds.

It can be seen from Figure 1B that the sprout length increases gradually with seed germination, and the increased rate of sprout length in the early stage (0–5 d) of germination is higher than that in the later stage (5–7 d) of germination. It is speculated that this may be related to the rapid transformation and synthesis of nutrients during the early stage of seed germination and tends to be balanced and stable in the later stage. The sprout length of the T_1_, T_2_, T_3_ and T_4_ treatment groups was significantly higher than that of the CK group (*p* < 0.05). There was no statistical difference in sprout length between the T_5_ group and CK group (*p* > 0.05), and at 7 days of germination, the sprout length of the CK group was higher than that of the T_5_ group. The results indicated that a certain concentration of exogenous conditions could promote the growth of sprouts, while a higher concentration could inhibit it.

### 3.2. Changes in Soluble Protein Content in TB Sprouts

The content of soluble protein can reflect the quality of plant metabolism. When plants are stressed, the content of soluble protein will increase greatly and act as osmotic regulator to alleviate the cell damage caused by stress [26]. Therefore, an increase in soluble protein can not only be used as a sign of stress, but also can alleviate the damage caused by stress. As shown in Figure 2, the content of protein in the sprout decreased significantly with the germination of TB seeds (*p* < 0.05). Compared with seeds, the content of soluble protein in the CK group decreased significantly by 57.72% after 3 days of germination (*p* < 0.05) and decreased gradually with the increase in germination time. The process of seed germination needs to consume a lot of nutrients to meet its own growth, so in this process, a large number of proteins in seeds are broken down into amino acids; with the extension of germination time, the sprouts are unable to carry out photosynthesis to provide the energy needed to synthesize proteins, so the proteins in the sprouts are gradually consumed. Among the five groups, the soluble protein content of the T_5_ group was the highest, which was 51.76% higher than that of CK group on the 5th day of germination, and 19.35%, 15.94% and 9.67% higher than the T_1_, T_2_ and T_3_ groups, respectively. It can be speculated that exogenous additives can stimulate the increase in soluble protein in TB sprouts to resist cell damage, and the content of soluble protein increases with the increase in the concentration of exogenous additives.

### 3.3. Changes in Soluble Sugar and Reducing Sugar Contents

Soluble sugar is the basis of plant metabolism and the main form of sugar metabolism, which not only meets the nutrients needed by plants but also plays a role in regulating osmotic pressure. As shown in Figure 3A, the content of soluble sugar in TB sprouts decreased gradually with seed germination. Under the condition of T_4_, the content of soluble sugar in TB sprouts germinated for 7 days was the lowest, at 2.24 g/100 g DW, which was significantly lower than that of seeds by 79.15% (*p* < 0.05). In the process of sugar metabolism, soluble sugar and starch are transformed and degraded by each other, which reduces the content of soluble sugar, while soluble sugar, such as sucrose, mainly comes from plant photosynthesis. TB sprouts will continue to consume the content of soluble sugar in sprouts to maintain their own growth. On the 3rd, 5th and 7th day of seed germination, the content of soluble sugar in the T_4_ group was lower than that in the CK group, and decreased by 15.48%, 17.87% and 53.05% compared with the CK group, respectively. During seed germination, osmotic regulation and temperature regulation are needed to achieve the purpose of initiation, which is the key to improving the seed germination rate and seedling uniformity. Combined with the results of the above appropriate concentrations of exogenous additives for promoting seed germination rate and sprout length, the treatment group consumed more soluble sugar during germination to regulate osmotic pressure and maintain its own growth than the CK group.

In the process of grain germination, macromolecular additives such as sugar, fat and protein in seeds are decomposed into small molecular additives, which are more easily absorbed and utilized, such as reducing sugars, fatty acids and amino acids, which participate in the metabolic process of matter and energy transformation. As shown in Figure 3B, the content of reducing sugar in TB seeds is the lowest, which is 3.70 g/100 g DW. With the germination of seeds, the content of reducing sugar in sprouts reached the highest at 7 days of germination and reached 19.12 g/100 g DW in the T_4_ group, which was significantly higher than that in the seed and CK groups (*p* < 0.05), and increased by 416.78% and 37.09%, respectively. This may be because the soluble sugar in TB is degraded after being converted into starch, and the starch is hydrolyzed into glucose, which provides energy for sprouting, and finally leads to an increase in reducing sugar content. On the other hand, it may be that the seed is stimulated by exogenous additives during germination, which leads to an increase in other nutrients, resulting in the transformation of reducing sugar from other nutrients.

### 3.4. Changes in Malondialdehyde (MDA) Content

MDA is the product of membrane lipid peroxidation. When plants are persecuted by stress, this will aggravate membrane peroxidation and lead to an increase in the content of MDA. MDA can bind to proteins and enzymes on the cell membrane and cause intramolecular and intermolecular cross-linking of proteins, thus destroying the structure and function of cell biofilms [27]. Therefore, the content of MDA can reflect the degree to which plants are persecuted by adversity. It can be seen from Figure 4 that the MDA content increased at first and then decreased with the seed germination, and reached the maximum at 5 days after seed germination. The MDA content in TB sprouts germinated for 5 days in the CK group was 4.63 mmol/g DW, which was 264.86% higher than that in seeds. During seed germination, the intensity of cell activity and metabolism increased, resulting in the increase in reactive oxygen species and membrane lipid peroxidation, so the content of MDA increased. It is also possible that during germination, the intracellular osmotic pressure increases, the cell absorbs water and expands, the integrity of the cell membrane is violated, and the increase in membrane permeability is more likely to produce more MDA. The decrease in MDA content after 5 days of germination may be related to the content of soluble protein in sprouts. Soluble protein initiates the enzyme defense system during seed imbibition and reduces the attack of reactive oxygen species on membrane lipids, so the MDA content decreases. On the 5th day of germination, the content of MDA in the T_3_ and T_5_ groups was the highest, which was 5.50 and 5.51 mmol/g DW, respectively, indicating that salt stress caused the most serious damage to cells during the seedling growth period. NaCl stress is a kind of abiotic stress, which increases the content of reactive oxygen species and MDA in plants [28]. The content of MDA in the T_2_ and T_4_ groups was relatively low, which was 4.16 and 4.23 mmol/g DW, respectively. It is suggested that the existence of L-Phe can inhibit the membrane lipid peroxidation of TB sprouts, thus alleviating the damage caused by salt stress, which may be related to the production of more antioxidant phenolics by L-Phe. The lower content of MDA in the T_1_ group may be related to the fact that CA can prevent the production of reactive oxygen species and reduce membrane lipid peroxidation [29].

### 3.5. TF and TP Contents before and after In Vitro Digestion

The effects of different treatments on the content of TF in TB sprouts are shown in Figure 5A. The content of TF increased with the extension of germination time, and the growth rate of TF content decreased gradually in the later stage of germination. On the 7th day of germination, the content of TF in the CK group was 5.76 g RE/100 g, which was 141.09% higher than that of seeds. The synthesis of flavonoids in plants is mainly through the phenylpropane metabolic pathway, which is affected by a series of related enzymes. In the early stage of germination, enzyme activity is very low, and with growth and metabolism, enzyme activity gradually increases and reaches the peak. Then, enzyme activity gradually decreased, so the synthesis rate of TF gradually slowed down in the later stage of germination. On the 3rd, 5th and 7th day of germination, the contents of TF in the T_4_ group were the highest, which were 5.24, 6.30 and 6.82 g RE/100 g, respectively, representing increases of 14.17%, 18.56% and 18.36%, respectively, compared with the CK. There was no statistical difference in the germination of TB in the T_1_, T_2_ and T_3_ groups (*p* > 0.05). The reason may be related to the metabolic balance in plants. Increasing the content of certain additives will accelerate biosynthesis, but the loss of other compounds will also increase.

The content of TF in each digestive period is shown in the line chart of Figure 5A. It can be seen that the content of TF decreased greatly after digestion. This may be related to the degradation of flavonoids by salivary amylase, pepsin and trypsin [30]. According to Sengul et al. [31], the content of TF in each *in vitro* digestion stage is mainly composed of food substrates containing bioactive compounds, because once flavonoids interact with other compounds, such as proteins, lipids, fibers, carbohydrates or minerals, their bioaccessibility changes. The content of TF increased with the progress of oral, gastric and intestinal digestion. Under the condition of T_4_, the contents of TF in oral, gastric and intestinal digestion were the highest, which were 1.32, 2.25 and 3.51 g RE/100 g, respectively, and the bioaccessibility was 51.50%. After intestinal digestion, the content of TF in each treatment group was significantly higher than that in the CK group (*p* < 0.05), indicating that exogenous additive treatment, especially under T_4_ treatment can effectively reduce the loss of TF caused by digestion, or can effectively improve the digestion and absorption of TF in the human body.

The effects of different treatments on the TP content of TB sprouts are shown in Figure 5B. After 7 days of germination, the TP content of the T_4_ and T_5_ groups reached the maximum, which was 17.04 and 16.94 mg GAE/g, respectively, representing increases of 20.96% and 20.23% compared with the CK, respectively. The contents of TP in the T_1_, T_2_ and T_3_ groups were higher than that in the CK group, but the differences have no statistical significance (*p* > 0.05). The results indicated that the effect of synergistic treatment of CA, L-Phe and NaCl on the TP content of TB sprouts was better than that of a single treatment.

As shown in the line chart of Figure 5B, after intestinal digestion, the content of TP in all groups was significantly lower than that before digestion (*p* < 0.05). In the T_4_ group, intestinal digestion was higher by 73.26% compared with that before digestion, and bioaccessibility was only 26.74%. This may be due to the autotrophication of phenolic compounds caused by dissolved oxygen and an alkaline environment during intestinal digestion, or because methanol used in chemical extraction before digestion can release phenolics and other bioactive components more effectively [31], especially when combined with ultrasound-assisted extraction [32]. After oral, gastric and intestinal digestion, the TP contents of the T_4_ group were the highest, which were 1.17, 3.19 and 4.56 mg GAE/g respectively, followed by the T_2_ and T_3_ groups, and lower in the T_1_ and T_5_ groups. It shows that L-Phe treatment was the most beneficial to the absorption of TP in TB sprouts, and the effect of CA was relatively lower than that of L-Phe and NaCl, and these results increased with the increase in its concentration. On the whole, the content of TP in all exogenous treatment groups was higher than that in the CK group.

### 3.6. DPPH, ABTS and FRAP Radical Scavenging Capacity before and after In Vitro Digestion

There are multiple antioxidant systems in plants, and the different modes of action of digestive activity *in vitro* cannot be explained by a single determination method, so when studying the antioxidant activity of additives, at least two or more methods should be selected to explain the antioxidant capacity [33].

The effects of different exogenous additives on the DPPH radical scavenging ability of germinated TB are shown in Table 2. After 7 days of germination, the antioxidant capacity of the T_4_ group was 174.74 μmol TE/g, which was significantly higher than that of other groups (*p* < 0.05), indicating that TB sprouts had the best scavenging effect on DPPH free radicals under T_4_ treatment. This may be related to the significant increase in the content of TF after L-Phe treatment. Our previous studies showed that the content of TF was positively proportional to the scavenging ability on DPPH free radicals [34], which was also consistent with the results of this experiment. The scavenging ability of seeds on DPPH free radicals decreased significantly after *in vitro* digestion (*p* < 0.05). After intestinal digestion, it was only 9.07 μmol TE/g, which was 83.61% lower than that before digestion, and the bioaccessibility was only 16.39%. Plant polyphenols are a kind of natural antioxidants, and their content affects the antioxidant capacity [35]. Phenolics are unstable in the weakly alkaline environment in the intestinal phase, and it is easy with macromolecules such as proteins to form complexes [36], which reduces the content of phenolics in digestive juice, resulting in a decrease in antioxidant capacity. After intestinal digestion, the free radical scavenging capacity on DPPH in the T_4_ group was 155.21 μmol TE/g, and the bioaccessibility was 88.82%. The antioxidant capacity of each digestive stage varies in different periods. The enhancement of antioxidant capacity in the intestinal stage can be attributed to the acidolysis of phenolic compounds such as glycoside flavonols under the action of gastric acid. Moreover, these phenolic compounds were transformed into more antioxidant aglycone [37], so the antioxidant activity was the strongest in the gastric and intestinal phases.

As shown in Table 2, the ABTS radical scavenging ability of the T_4_ group was the highest after 7 days of germination, at 278.99 μmol TE/g, which was 27.80% and 257.55% higher than that of the CK group and seeds, respectively. The results showed that exogenous additives could increase the antioxidant capacity of TB to some extent. Although there was no statistical difference in ABTS radical scavenging ability between the T_2_ and T_4_ groups 7 days after germination, the antioxidant capacity of the T_2_ group was 267.22 μmol TE/g, which was 4.22% lower than that of the T_4_ group. Therefore, the effect of L-Phe alone on the antioxidant capacity of TB was lower than that of synergistic promotion, which may be related to the activity of different antioxidants stimulated by different exogenous additives. After *in vitro* digestion, the scavenging ability on ABTS free radicals in sprouts was significantly lower than that before digestion (*p* < 0.05), and the decreasing trend was the most obvious after oral and gastric digestion. After oral and gastric digestion, the scavenging capacity on ABTS in the CK group was 54.95 and 70.59 μmol TE/g, respectively, which was 74.83% and 67.66% lower than that before digestion, respectively. After intestinal digestion, the scavenging ability on ABTS free radicals in sprouts increased significantly compared with oral and gastric phases (*p* < 0.05). After intestinal digestion, the antioxidant activity of the T_4_ group reached 196.26 μmol TE/g, which was 29.65% lower than that before digestion, and the bioaccessibility was 70.35%. This may be related to an increase in polyphenols after intestinal digestion and the conversion of phenolics into simpler compounds in the intestinal phase.

As shown in Table 2, the FRAPs of the T_1_, T_2_ and T_4_ groups were significantly higher than those of other treatment groups, which increased by 23.78%, 31.11% and 37.78% compared with the CK group (*p* < 0.05), respectively. The result indicated that CA and L-Phe treatment could increase the FRAP in sprouts to some extent, but a higher concentration (T_5_) could inhibit the antioxidant activity. According to the research results presented in Section 3.5 above, the contents of TP and TF in TB sprouts will increase significantly after germination (*p* < 0.05), so phenolics as antioxidants will play a strong antioxidant role. In addition, the physical properties of flavonoids determine their interaction with the cell membrane, because hydrophobic flavonoids can be deeply embedded in the cell membrane, where they can affect the membrane fluidity and disrupt the oxidation chain reaction [38]. Since plant chemicals and their combinations also have certain effects on the effective structure and physical properties of the cell membrane [39], the effects of the binding of polysaccharides and proteins to the cell membrane cannot be ruled out. After *in vitro* digestion, the FRAP decreased significantly. After intestinal digestion, the FRAP of the T_4_ group was still the highest, at 28.14 μmol TE/g, which was 10.79%, 4.62%, 8.70% and 16.40% higher than those of the T_1_, T_2_, T_3_ and T_5_ groups, respectively. The results showed that exogenous treatment in the T_4_ group could also increase the FRAP of TB sprouts.

In the T_4_ group, DPPH, ABTS radical scavenging ability and FRAP of TB sprouts were 174.74, 278.99 and 45.25 μmol TE/g after 7 days of germination, respectively. The values after intestinal digestion were 155.21, 196.26 and 28.14 μmol TE/g, respectively. The bioaccessibility was 88.82%, 70.35% and 62.19%, respectively. The results showed that the antioxidant capacity on the ABTS free radical scavenging system was the strongest, but the bioaccessibility of TB sprouts was the highest when studying the antioxidant ability of TB sprouts on the DPPH free radical scavenging system. That is, after the treatment of T_4_ exogenous conditions, the human body can greatly improve the scavenging ability on DPPH free radicals.

### 3.7. Principal Component Analysis

The TP and TF (before and after the oral and gastrointestinal digestion) are standardized in a two-dimensional system (Biplot graph) through principal component analysis (PCA) to establish the correlation among the parameters and the different treatments (Figure 6A). A total of 99.40% of the variation was explained by the first principal component (PC1) (93.30%) and second principal component (PC2) (6.10%). The germination time was positively correlated with these indexes, and the effect of the T_4_ treatment group on these parameters was more obvious. Likewise, TF and TP presented a high and positive correlation with each other, except for the oral stage TP. Compared with the CK group, exogenous conditioned treatment (T_4_) had a strong correlation with TP after intestinal digestion and a high correlation with TF in the oral, gastric and intestinal phases. Combined with the above analysis, T_4_ treatment can not only increase the content of phenolics in TB sprouts before digestion but also alleviate the nutrient loss caused by digestive juice. This result further confirms our speculation that exogenous additives can stimulate the biosynthesis of endogenous compounds and then play a role in plants.

As shown in Figure 6B, the PCA showed three principal components (PC), and the first two PC explain 97.00% of the total variability. PC1 showed the highest contribution (86.50%), and this is due to TP, TF, DPPH, ABTS and FRAP. In PC2, the highest contribution (10.50%) came from the MDA. There are statistical differences between the five groups of the first two principal components. TP and TF were positively proportional to DPPH, ABTS and FRAP, and had a strong correlation with TB sprouts germinated for 7 days (CK group and T_4_ group), and the T_4_ group had the strongest correlation with it. The results indicated that both germination and T_4_ treatment promoted the increase in phenolics and then improved the antioxidant capacity, and T_4_ treatment had a more significant impact on this effect. There was a strong correlation between MDA and the CK group after germination for 3 and 7 days, indicating that the stress degree of T_4_ treatment on TB seeds was lower than that of germination alone, or that T_4_ treatment could alleviate the stress caused by germination. In addition, soluble protein and soluble sugar were negatively proportional to other indexes, which explained that TB seed germination could not produce energy and provide a source through photosynthesis under the condition of avoiding light, so it could only be consumed into small molecular amino acids and reducing sugars by the sprouts to provide nutrition for germination.

To sum up, the contents of TF and TP before digestion were significantly correlated with TB sprouts germinated for 3 days, while the contents of TF and TP after digestion were significantly correlated with TB sprouts germinated for 7 days, except for TP in gastric stage, which indicated that although the contents of TF and TP in TB sprouts were synthesized in the early stage of germination, the TF in TB sprouts germinated for 7 days were more easily digested and absorbed by the human body. Combined with the information in Table 2, the DPPH, ABTS and FRAP of TB sprouts germinated for 7 days increased gradually from oral to intestine after *in vitro* digestion, which was related to the fact that TF and TP were more easily digested when TB germinated for 7 days. The antioxidant capacity of TB sprouts before digestion was significantly correlated with that of TB sprouts germinated for 7 days, which was mainly related to the high contents of TF and TP.

## 4. Discussion

NaCl stress can induce phenolic acid accumulation and synthesis of some antioxidant enzymes in plants, such as superoxide dismutase (SOD) [28] and ascorbate peroxidase (APX) [40]. The action of these enzymes can effectively scavenge reactive oxygen species and free radicals, thus maintaining the normal growth and development of plants. In addition, some studies have shown that the acquisition of plant salt tolerance may also be a consequence of improving resistance to oxidative stress, via increased APX activity and a change in the profile of fatty acids [41]. The results of this study can also confirm that NaCl stimulation improved the germination characteristics and antioxidant capacity of Tartary buckwheat. The growth and metabolism of plants are inseparable from the catalytic degradation of enzymes. Therefore, the activities of corresponding enzymes in germinated Tartary buckwheat under salt stress should continue to be studied, so as to establish the correlation between phenolics, antioxidant enzymes and antioxidant capacity.

Phenolic acids are a kind of important secondary metabolites widely distributed in plants, which mainly exist in the form of ester bonds, glycosidic bonds and other substances (including proteins, flavonoids, monosaccharides, etc.) [42]. CA as a natural compound with a hydroxyphenylenoic acid structure can be combined with other active substances to form esters or amides to obtain new derivatives with anticancer, antioxidant and anti-inflammatory activities, such as caffeic acid phenethyl ester (CAPE) [43]. However, the increase in antioxidant activity in this paper cannot confirm the addition effect of CA combined with other active components into new substances after Tartary buckwheat germination, so further study is needed.

L-Phe is not only the intermediate product of plant phenylpropane metabolism but also the precursor of flavonoid biosynthesis. Adding exogenous L-Phe can effectively enrich flavonoids, increase the content of polyphenols and other nutrients and enhance the effect of antioxidation [10]. Combined with previous experience and our previous treatment, it was found that a single exogenous condition treatment could significantly increase the content of the main active components and antioxidant capacity of Tartary buckwheat sprouts. This paper boldly combines NaCl, CA and L-Phe, hoping that they can improve the quality of Tartary buckwheat sprouts in different aspects and finally stimulate the potential function of Tartary buckwheat sprouts. However, we should continue to study the synergistic promotion mechanism of the three.

## 5. Conclusions

The purpose of this study was to explore the effects of CA, L-Phe and NaCl on the main active components, antioxidant capacity and *in vitro* digestion of Tartary buckwheat after germination. The results showed that TB seed germination under the condition of avoiding light could lead to a decrease in soluble protein and soluble sugar, an increase in the contents of TF, TP and MDA, and an increase in antioxidant activity. Exogenous condition treatment can alleviate the decrease in soluble components caused by germination and increase phenolic substances and antioxidant capacity, among which CA 17 mg/L, L-Phe 2.7 mmol/L and NaCl 2.7 mmol/L cooperative treatment (T_4_) is the best. In addition, *in vitro* digestion will lead to a decrease in phenolics content and antioxidant capacity. After oral, gastric and intestinal digestion, the bioaccessibility of DPPH, ABTS and FRAP reached 88.82%, 70.35% and 62.19%, respectively. Therefore, this study indicated that the treatment with appropriate concentrations of exogenous additives can be used as an effective strategy for improving Tartary buckwheat germination and the active components of Tartary buckwheat, and provide a certain reference value for the development and utilization of functional food. Although exogenous additives can increase the bioaccessibility of TB after digestion, there is no in-depth research on its mechanism, so it is necessary to study the mitigation of exogenous additives on the loss of related components after digestion at the genetic or molecular level.

## Figures and Tables

**Figure 1 foods-11-03682-f001:**
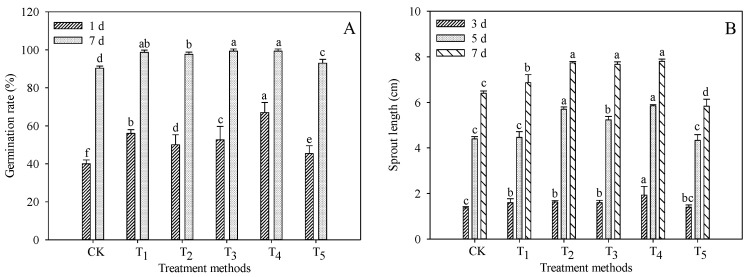
Effect of exogenous additives on the growth of TB sprouts. (**A**) Germination rate (%); (**B**) Sprout length (cm). Different lowercase letters indicate significant statistical difference (*p* < 0.05).

**Figure 2 foods-11-03682-f002:**
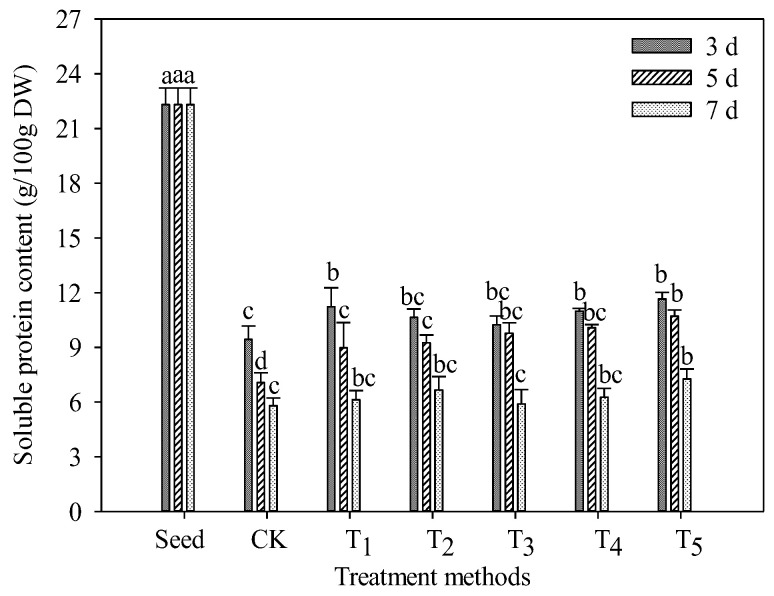
Effect of exogenous additives on the soluble protein content of TB sprouts. Different lowercase letters indicate significant statistical difference (*p* < 0.05).

**Figure 3 foods-11-03682-f003:**
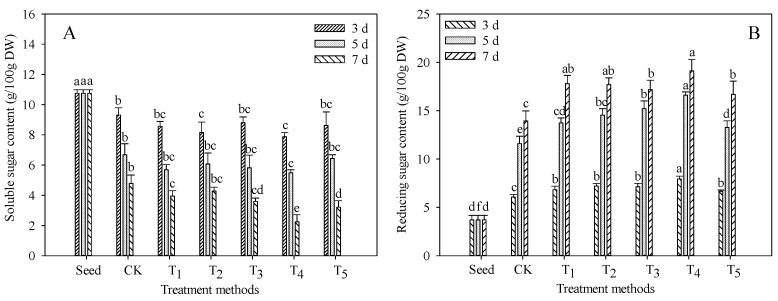
Effect of exogenous additives on soluble sugar and reducing sugar contents of TB sprouts. (**A**) Soluble sugar content (g/100 g DW); (**B**) Reducing sugar content (g/100 g DW). Different lowercase letters indicate significant statistical difference (*p* < 0.05).

**Figure 4 foods-11-03682-f004:**
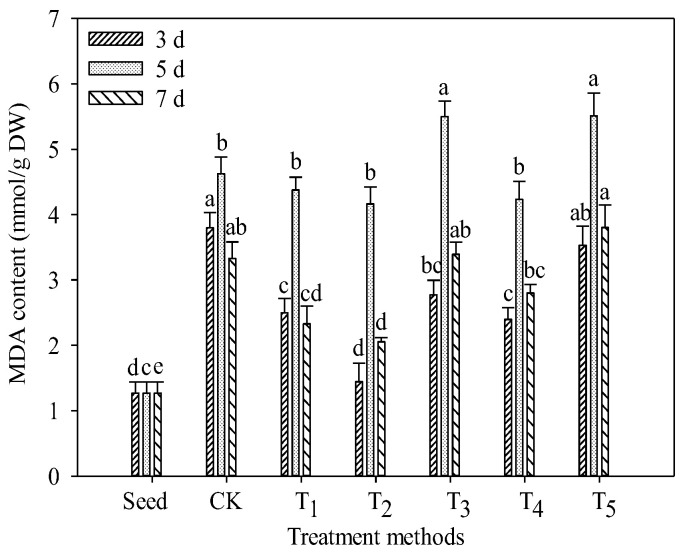
Effect of exogenous additives on MDA content in TB sprouts. Different lowercase letters indicate significant statistical difference (*p* < 0.05).

**Figure 5 foods-11-03682-f005:**
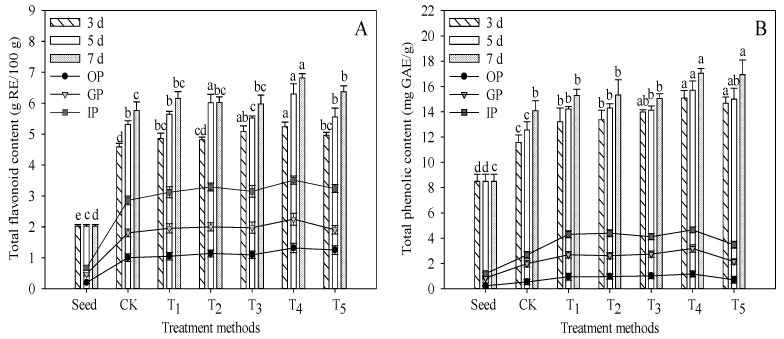
Effects of *in vitro* digestion on TP and TF contents of TB sprouts. OP, oral phase; GP, gastric phase; IP, intestinal phase. (**A**) Total flavonoid content (g RE/100 g); (**B**) Total phenolic content (mg GAE/g). Different lowercase letters indicate significant statistical difference (*p* < 0.05).

**Figure 6 foods-11-03682-f006:**
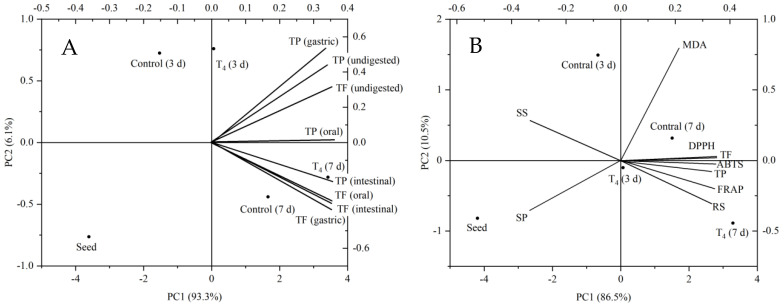
Component plot of main active components (PCA), TF, TP and antioxidant activity before and after digestion on TB seed and sprouts. SS, soluble sugar; SP, soluble protein; RS, reducing sugar. (**A**) PCA of TF and TP before and after digestion on TB sprouts; (**B**) PCA of antioxidant activity and active components on TB sprouts.

**Table 1 foods-11-03682-t001:** Concentration of exogenous additives and germination conditions of TB sprouts.

Treatment Methods	Treatment Conditions		
	CA (mg/L)	L-Phe (mmol/L)	NaCl (mmol/L)
CK	0	0	0
T_1_	17	0	0
T_2_	0	2.7	0
T_3_	0	0	2.7
T_4_	17	2.7	2.7
T_5_	30	2.7	2.7

**Table 2 foods-11-03682-t002:** Antioxidant capacity (μmol TE/g) before and after in vitro digestion.

	Chemical Extraction			Oral Phase			Gastric Phase			Intestinal Phase			Bioaccessibility/%		
	DPPH	ABTS	FRAP	DPPH	ABTS	FRAP	DPPH	ABTS	FRAP	DPPH	ABTS	FRAP	DPPH	ABTS	FRAP
Seed	55.37 ± 1.51 ^d^	78.03 ± 20.65 ^d^	12.35 ± 0.52 ^f^	3.09 ± 0.31 ^e^	6.57 ± 0.29 ^e^	3.06 ± 0.16 ^f^	8.98 ± 0.25 ^e^	12.64 ± 0.33 ^f^	5.87 ± 0.12 ^e^	9.07 ± 0.55 ^f^	16.53 ± 0.10 ^f^	6.54 ± 0.21 ^f^	16.39	21.18	52.96
T_1_	159.37 ± 6.24 ^bc^	235.64 ± 5.97 ^c^	40.65 ± 1.80 ^c^	42.11 ± 1.49 ^b^	68.97 ± 0.69 ^b^	12.09 ± 0.38 ^b^	92.63 ± 3.22 ^c^	89.41 ± 1.49 ^c^	17.96 ± 0.71 ^c^	123.55 ± 8.23 ^d^	183.91 ± 2.45 ^c^	25.40 ± 0.25 ^c^	77.53	78.05	62.48
T_2_	166.09 ± 5.26 ^a^	267.22 ± 6.19 ^ab^	43.06 ± 1.90 ^b^	42.92 ± 0.69 ^b^	70.18 ± 0.66 ^ab^	12.42 ± 0.36 ^b^	99.06 ± 3.90 ^b^	94.01 ± 2.11 ^b^	19.20 ± 0.27 ^b^	142.19 ± 5.08 ^b^	189.34 ± 1.47 ^b^	26.90 ± 0.58 ^b^	85.61	70.85	62.46
T_3_	160.81 ± 7.83 ^b^	244.31 ± 10.19 ^bc^	37.72 ± 0.72 ^d^	39.46 ± 1.08 ^c^	67.63 ± 1.89 ^c^	10.70 ± 0.50 ^d^	103.04 ± 4.91 ^b^	89.32 ± 2.02 ^c^	18.58 ± 1.01 ^c^	131.86 ± 5.40 ^c^	180.93 ± 4.31 ^c^	25.89 ± 0.34 ^c^	82.00	74.06	68.63
T_4_	174.74 ± 3.21 ^a^	278.99 ± 6.50 ^a^	45.25 ± 2.08 ^a^	51.05 ± 1.62 ^a^	73.43 ± 1.55 ^a^	13.44 ± 0.38 ^a^	125.99 ± 6.72 ^a^	109.54 ± 1.62 ^a^	20.27 ± 0.57 ^a^	155.21 ± 2.72 ^a^	196.26 ± 4.29 ^a^	28.14 ± 0.60 ^a^	88.82	70.35	62.19
T_5_	159.94 ± 3.17 ^c^	236.87 ± 22.48 ^c^	37.75 ± 2.16 ^d^	44.13 ± 1.65 ^b^	67.49 ± 0.84 ^c^	11.20 ± 0.62 ^c^	98.60 ± 4.87 ^bc^	84.39 ± 3.59 ^d^	19.12 ± 0.61 ^b^	141.74 ± 7.86 ^b^	167.37 ± 2.54 ^d^	24.17 ± 0.58 ^d^	88.62	70.66	64.04
CK	147.07 ± 5.28 ^c^	218.30 ± 17.14 ^c^	32.84 ± 1.07 ^e^	34.50 ± 1.97 ^d^	54.95 ± 0.56 ^d^	9.89 ± 0.24 ^e^	79.78 ± 5.28 ^d^	70.59 ± 2.11 ^e^	15.89 ± 0.54 ^d^	107.39 ± 5.73 ^e^	148.92 ± 6.76 ^e^	21.24 ± 0.76 ^e^	73.02	68.22	64.66

Means ± SD. Different lowercase letters per column indicate significant statistical difference (*p* < 0.05).

## Data Availability

The data generated during the present study are available from the corresponding author upon reasonable request.

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
