# Peer review of "Effects of Exogenous Caffeic Acid, L-Phenylalanine and NaCl Treatments on Main Active Components Content and In Vitro Digestion of Germinated Tartary Buckwheat"

_foods, 2022, doi:10.3390/foods11223682_

Round 1

Reviewer 1 Report

The authors have investigated the effects of exogenous additives treatments (caffeic acid, L-phenylalanine, NaCl) on germination of Tartary buckwheat (TB) and studied the improvement in main active components contents and antioxidant activities before and after in vitro digestion of germinated (TB). The results are good and well explained. However, the manuscript needs revisions in the following points: 

  1. In abstract, various terms  (CK group, T4 group ) are mentioned without any introduction does not give a better understanding to readers. Hence, a modification in abstract is needed. 
  2. In a similar way TF and TP are used as an abreaction throughout the manuscript without defining them. Abreactions must be defined first time when they are used.  
  3. A critical grammar check is required. For example respectively is used frequently for following a standard procedure.  
  4. Line 108, equations should be written in italic. 
  5. In line 198, mention Figure 1B as Figure 1 (B) for better visibility. 
  6. Check line 129 and 113, use consistency in writing gram or g. 
  7. Quality of Figure 4 should be improved. 
  8. The number of references reported are reasonable and in order. I suggest to include few more recent references, if any. 

In conclusion, I recommend that the author to revise the manuscript in the light of above suggestions.  I recommend a minor revision.

Author Response

Point 1. In abstract, various terms (CK group, T4 group ) are mentioned without any introduction does not give a better understanding to readers. Hence, a modification in abstract is needed.

Response: Thank you for your correction. We are sorry for the trouble caused by our negligence to your smooth review. We have replaced the relevant introduction in the summary. Thank you again for your guidance.

See line 12-16: “Compared with natural growth group, T4 group (CA 17 mg/L, L-Phe 2.7 mmol/L, NaCl 2.7 mmol/L) treatment increased the germination rate (67.50%)...”

Point 2. In a similar way TF and TP are used as an abreaction throughout the manuscript without defining them. Abreactions must be defined first time when they are used.

Response: Thank you very much. We once again express our apologies to you. Due to our negligence in writing and reviewing, the full name explanation of TF and TP is missing in the full manuscript. These two words appear widely in the full article, causing trouble for your reading. This is a very serious mistake. Once again, we would like to express my apology and thanks to you. See line 47.

Point 3. A critical grammar check is required. For example respectively is used frequently for following a standard procedure.

Response: Thank you for your valuable and thoughtful comments. We have carefully checked and improved the English writing in the revised manuscript.

Point 4. Line 108, equations should be written in italic.

Response: Thank you for your correction. We have revised it in our manuscript. See line 109, 130, 138.

Point 5. In line 198, mention Figure 1B as Figure 1 (B) for better visibility.

Response: First of all, thank you very much for your suggestion, and we fully agree with you, but there is a requirement for the use of this citation in the ‘Style Guide’ of the Foods journal, so we cannot make any changes to the suggestion. Thank you again. The excerpts of the examples are as follows:

“7. Figures, Tables, and Data:

Any figures, tables, supplementary information, etc., must be cited in the main text of the document, e.g.,

‘The data are shown in Table 3.’

‘This case is depicted in Figure 3d.’”

Point 6. Check line 129 and 113, use consistency in writing gram or g.

Response: Thank you for your correction. We have checked and corrected this, and in this article we have replaced gram with g. See line 159, 165, 188.

Point 7. Quality of Figure 4 should be improved.

Response: Thank you very much for your professional guidance. We have re-uploaded this image to make it clearer in resolution and more suitable in size. In addition, we have re-uploaded other low-quality figure in the revised article. Once again, we would like to express our gratitude to you.

Point 8. The number of references reported are reasonable and in order. I suggest to include few more recent references, if any.

Response: Thank you for your careful guidance. We have tried our best to replace the older references to make the article more meaningful for discussion. See the “References”.

Reviewer 2 Report

Dear author

I report a review result about ‘Effects of exogenous caffeic acid, L-phenylalanine and NaCl treatments

on main active components content and in vitro digestion of germinated

Tartary buckwheat’. There are some revision points. Please confirm the item which I pointed out. Please inform it if there are my deficiency.

Best regard.

1. L33:  Addition of reference

I think that the reference should be add about rich in flavonoids, vitamins, trace elements and various essential amino acids with antioxidant activity.

2. L44: Addition of explanation to the first abbreviation go out

Please add explanation to an abbreviation of the first to go out.

TF (total flavonoid)

TP (total polyphenol)

3. Figure 5: It is hard to understand the distinction between and .

It is hard to understand the distinction of markers between OP: and IP. Please improve that the change marker size or change marker shape

4. Correlation among antioxidant characteristics

Please describe about the correlation among antioxidant characteristics, DPPH, ABTS, FRAP, TF and TP before and after digestion.

Author Response

Point 1. L33: Addition of reference

I think that the reference should be add about rich in flavonoids, vitamins, trace elements and various essential amino acids with antioxidant activity.

Response: Thank you for your guidance. We very much agree with your opinion, it is a great suggestion and it is very necessary to add references in this place. We have added relevant references here, once again express our thanks to you. See line 36.

Point 2. L44: Addition of explanation to the first abbreviation go out

Please add explanation to an abbreviation of the first to go out.

TF (total flavonoid)

TP (total polyphenol)

Response: Thank you for your correction, and we would like to apologize for our carelessness, because this mistake has troubled your reading. As answered by the first reviewer, we have corrected this mistake and thank you again. See line 47.

Point 3. Figure 5: It is hard to understand the distinction between ● and ▼.

It is hard to understand the distinction of markers between OP:● and IP▼. Please improve that the change marker size or change marker shape

Response: First of all, we are very sorry for the reading disorder caused by the mistakes in drawing. Secondly, we are very grateful for your valuable comments, which we have revised carefully. And finally we would like to express our gratitude to you again.

Point 4. Correlation among antioxidant characteristics

Please describe about the correlation among antioxidant characteristics, DPPH, ABTS, FRAP, TF and TP before and after digestion.

Response: Thank you very much for your valuable advice. In the process of writing, we neglected the description of the correlation between antioxidant capacity, total flavonoids and total phenolics content of Tartary buckwheat sprouts before and after digestion. In view of this problem, we synthesize the previous separate explanations of total antioxidant capacity, total flavonoids and total phenolics, and roughly add a description of their correlation at the end of the PCA analysis. We hope you can have an in-depth communication with us again on this issue. Finally, we would like to express our gratitude to you again.

See line 518-529: “To sum up, the contents of TF ... which was mainly related to the high contents of TF and TP.”

Reviewer 3 Report

The manuscript entitled “Effects of exogenous caffeic acid, L-phenylalanine and NaCl treatments on main active components content and in vitro digestion of germinated Tartary buckwheat” describes the effect of different treatments on seed germination, sprouts length, soluble proteins, sugars (soluble and reducing), MDA, total phenolics content, total flavonoids content, and antioxidant capacity. The authors also analyzed bioaccessibility by applying in vitro digestion. The manuscript is well structured and quite interesting, but some changes must be done. Description of methods in sub-sections 2.3., 2.4., 2.5. and 2.6. should be more detailed in terms of extraction of analyzed compounds from sprouts, the volume of sample and reagent, and parameters of incubations (temperature, time, etc.). The authors used UV-VIS spectrophotometer for the analysis, please provide information about the model and vendor.  

Author Response

Point 1. Description of methods in sub-sections 2.3., 2.4., 2.5. and 2.6. should be more detailed in terms of extraction of analyzed compounds from sprouts, the volume of sample and reagent, and parameters of incubations (temperature, time, etc.).

Response: Thank you very much for your guidance. We have made a detailed supplement to the lack of this part. If there are any other mistakes in the supplement, we hope you will not hesitate to give us your advice. Thank you again for your guidance.

See line 100-103: “0.2 g fresh TB sprouts ... and the supernatant was protein extract.”

Line 104-109: “The 0.1 mL protein extract ... And bovine serum protein was used as a standard substance to make a standard curve”

Line 114-124: “0.5 g TB sprout powder ... The supernatant was used for the determination of soluble sugar and reducing sugar.”

Line 126-129: “2 mL supernatant ... After the mixture was cooled to room temperature in ice water and the absorbance was determined at 620 nm.”

Line 134-137: “2 mL supernatant ... Finally, the absorbance was determined at 540 nm.”

Line 142-148: “0.3 g fresh TB sprouts ... and determined the absorbance at 532, 660, and 450 nm.”

Point 2. The authors used UV-VIS spectrophotometer for the analysis, please provide information about the model and vendor.

Response: Thank you for your careful correction, this aspect is indeed easy for us to ignore. We have carefully examined this part of the content. I hope you can once again propose amendments to me, and finally express our thanks to you again. See line 107.

Others:

  1. Line 16-17, 21-22: Because we changed the expression of T4 and CK, the number of words in the abstract exceeded 200 words, so we revised these two sentences again so that the number of words was within the required range.
  2. We merged 2.4 and 2.5. Because the extraction steps of crude sugar are the same for the determination of reducing sugar and soluble sugar, in order to make the description of the article more concise and reasonable, we modified this part of the content.

Round 2

Reviewer 3 Report

The authors are accepted all my suggestions thus the manuscript can be accepted in the present form.